# DIA: Diffusion based Inverse Network Attack on Collaborative Inference

## Abstract

With the continuous expansion of neural networks in size and depth, and the growing popularity of machine learning as a service, collaborative inference systems present a promising approach for deploying models in resource-constrained computing environments. However, as the deployment of these systems gains traction, evaluating their privacy and security has become a critical issue. Towards this goal, this paper introduces a diffusion-based inverse network attack, named DIA, for collaborative inference systems that uses a novel feature map awareness conditioning mechanism to guide the diffusion model. Compared to prior approaches, our extensive empirical results demonstrate that the proposed attack achieves an average improvement of $29\%$, $20\%$, $30\%$ in terms of SSIM, PSNR, and MSE when applied to convolutional neural networks (CNN), $18\%$, $17\%$, $61\%$ to ResNet models, and $55\%$, $54\%$, $84\%$ to Vision transformers (ViTs). Our results identify the significant vulnerability of ViTs and analyze the potential sources of this vulnerability. Based on our analysis, we raise caution regarding the deployment of transformer-based models in collaborative inference systems, emphasizing the need for careful consideration regarding the security of such models in collaborative settings.

## 1 Introduction

With the rapid development of deep learning, especially with large language models (LLMs), the application scenarios of machine learning as a service (MLaaS) are becoming increasingly diverse (Hunt et al., 2018; Noels et al., 2023). These widespread applications have driven the exploration of deploying large models on constrained computation and storage resources such as Internet-of-Things (IoT) and edge devices, leading to the emergence of collaborative inference as a prevalent method (Hauswald et al., 2014; Teerapittayanon et al., 2017; Kang et al., 2017; Ko et al., 2018; Eshratifar et al., 2019; Zhang et al., 2023). In a collaborative inference system, a model is divided into multiple segments, with different devices performing inference on distinct segments of the model.

In the most common, two-party collaborative inference architecture, a model is split into two parts: the client retains the initial segment, while the computationally capable server handles the latter half. Upon completing the inference of the model's first portion, the client forwards the intermediate feature maps to the server, which then completes the computation and returns the final results. Typically, the model's initial segment contains fewer layers ensuring minimal computational overhead. In contrast, the latter segment is more computationally intensive, often encompassing full-connection layers. This paradigm effectively alleviates the computational burden on the client.

From the perspective of data privacy for the end client, it may appear secure as the server does not have direct access to user data. However, recent studies (Zhang et al., 2023; Li et al., 2022; Yin et al., 2023; He et al., 2019) suggest that given access to query the client's model, the adversary can train an inverse model based on intermediate feature maps, potentially enabling the reconstruction of input data. Such inverse network attacks mainly relied on transposed convolutional neural networks as the attack model. These networks commonly employ transposed convolution layers as a means of inverting normal convolutions, yielding effective attack results.

However, as neural networks grow in depth, the use of more nonlinear layers introduces increasing resilience of inversion network attack (He et al., 2019; Zhang et al., 2023; Yin et al., 2023), mak-

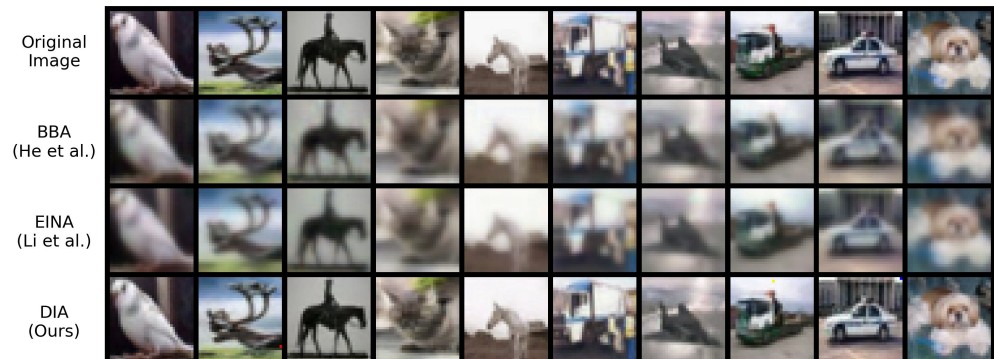

Figure 1: Reconstructed inputs comparison of prior approaches and our method on ReLU22 of CNN

ing it more challenging to reconstruct using transposed convolution layers alone. To enhance the reconstruction, a more proficient inverse model with generative capabilities is required. Conditional diffusion models, such as stable diffusion (Rombach et al., 2022), utilizing cross-attention as a condition mechanism demonstrate the superior capability in generating diverse and high-fidelity data across various domains, including images, videos, music, and audio (Dhariwal & Nichol, 2021; Ho et al., 2022; Levkovitch et al., 2022; Poole et al., 2022; Mittal et al., 2021), making them potentially suitable for the task of generating lost information based on intermediate feature maps. Thus motivated, in this paper, we present a diffusion-based inverse attack for reconstructing input data. A key feature of the proposed attack is the introduction of a novel feature map awareness conditioning mechanism with a companion network specifically designed for inverse network attacks. We use the Structural Similarity Index (SSIM), Peak Signal-to-Noise Ratio (PSNR), and Mean Squared Error (MSE) (He et al., 2019; Yin et al., 2023) to evaluate inverse network attacks. In comparison to previous attacks, as shown in Figure 1, the extensive empirical results show that the proposed attack achieves an average improvement of 29%, 20%, 30% in terms of SSIM, PSNR, and MSE when applied to convolutional neural networks (CNN) and 18%, 17%, 61% to ResNet models. Moreover, as the demand for sophisticated MLaaS continues to grow, it becomes essential to consider not only CNNs but also prevalent transformer-based models such as Vision Transformers (ViTs) (Dosovitskiy et al., 2020; Touvron et al., 2021) and LLMs (Vaswani et al., 2017; Touvron et al., 2023; OpenAI, 2023) within collaborative inference systems. The privacy of the transformer-based models has remained relatively unexplored in the context of collaborative inference. In this paper, we apply the proposed diffusion-based attack on the ViT and uncover its notable vulnerability compared to CNNs and ResNet.

We summarize our contributions as follows:

- We present a diffusion-based inverse network attack on collaborative inference systems. In comparison to prior attack methods, our empirical results demonstrate a significant improvement in reconstructed results, as evidenced by higher SSIM, PSNR, and lower MSE.

- We present a novel feature map awareness conditioning mechanism that uses a companion network that is specifically designed to enhance inverse network attacks. Our experimental results indicate the better performance of this conditioning method when compared to the cross-attention conditioning approach.

- By employing the diffusion-based attack, we investigate the vulnerability of the transformer-based model, ViTs, in collaborative inference systems. Our empirical findings reveal that ViTs are significantly more vulnerable than CNNs and ResNet models. We analyze this vulnerability from two perspectives and raise caution accordingly regarding the deployment of transformer-based models in collaborative inference systems.

## 2 Background

### 2.1 Metrics for Evaluating Inverse Network Attack

To evaluate the attack results, the prior method (He et al., 2019) applies two metrics, PSNR and SSIM. PSNR quantifies the pixel-level recovery quality of an image, with higher PSNR values indicating better image reconstruction quality. On the other hand, SSIM (Wang et al., 2004) measures the human perceptual similarity between two images by considering factors like luminance, contrast, and structural aspects. SSIM values range between 0 and 1, where 0 indicates minimal similarity and 1.0 represents maximum similarity. Moreover, another prior method, Ginver (Yin et al., 2023) employs Mean Squared Error (MSE) to measure pixel-wise differences between two images, with a smaller MSE indicating higher similarity. In this paper, for a comprehensive evaluation, we utilize all three metrics to assess the proposed attack.

### 2.2 Denoising Diffusion Probabilistic Model and Conditional Diffusion Model

Denoising Diffusion Probabilistic Model (DDPM) (Ho et al., 2020) have demonstrated state-of-the-art results in image (Song et al., 2020), speech synthesis (Chen et al., 2020), and time series forecasting (Rasul et al., 2021). The diffusion models have two processes: the forward and reverse processes. In the forward process, noise is incrementally introduced into the images, transforming them into Gaussian distributions. In contrast, the reverse process is an iterative denoising procedure that initiates from a sampled noise. To execute the reverse process, a UNet is trained to predict the added noise at each time step, which is subsequently removed using denoising operations. The central concept behind UNet training lies in predicting the distribution of the introduced noise across various time steps.

As for conditional diffusion, an encoder based on the modality of the condition is usually utilized to transform condition information into an embedding. This embedding is then integrated into the UNet model using a conditioning approach. A prevalent conditioning approach is cross-attention (Rombach et al., 2022), known for its effectiveness in generating diverse and high-fidelity data across various domains. The success of diffusion models, particularly conditional diffusion models, inspires our exploration of designing a diffusion-based inverse network model for reconstructing the input data.

## 3 The Proposed Diffusion Based Inverse Network Attack

### 3.1 Notations

In this paper, we refer to the first part of the model on the data owner-side with the notation $M_1$. The diffusion process is characterized by $q$ and $p$ representing the forward and backward processes, respectively. Within this context, $x_t$ denotes the noisy image at the time step $t$ and $\epsilon$ represents the noise. Additionally, $y$ denotes the intermediate feature map that the adversary can access. $T$ represents the predetermined number of steps used in the Gaussian diffusion process.

### 3.2 The Threat Model

Similar to prior works (He et al., 2019; Yin et al., 2023; Zhang et al., 2023), we focus on the most common collaborative inference paradigm, specifically the two-party system. The proposed approach is performed in a black-box setting: the server has no knowledge about the data owner or client's model $M_1$, including its architecture and parameters. During collaborative inference, the curious server attempts to retrieve the client's private input $x$ from the received $M_1(x)$. We assume the server can query the model to obtain the corresponding intermediate feature output to train an inversion model.

### 3.3 OUR METHOD

#### 3.3.1 FORWARD DIFFUSION PROCESS AND TRAINING

Following the regular diffusion model (Ho et al., 2020), the proposed approach defines a forward Markovian diffusion process denoted as $q$. This process involves iteratively introducing Gaussian noise to the input image $x$ over $T$ iterations:

$$q(x_{1:T}|x_0) = \Pi_{t=1}^{T} q(x_t|x_{t-1}), \tag{1}$$

$$q(x_t|x_{x-1}) = \mathcal{N}(x_t|\sqrt{1-\beta_t}x_{t-1}, \beta_t I), \tag{2}$$

where $\beta_t$ represents the variance of the noise added at time step $t$, ranging between 0 and 1. This variance gradually increases, ensuring the acquisition of isotropic Gaussian noise after $T$ iterations. Given the initial image $x_0$, the distribution of $x_t$ can be expressed as:

$$q(x_t|x_0) = \mathcal{N}(x_t|\sqrt{\gamma_t}x_0, (1-\gamma_t)I), \tag{3}$$

where $\gamma_t$ is defined as $\Pi_{t=1}^{t}(1-\beta_t)$.

As shown in Algorithm 1, in the training process that aims at optimizing the conditional denoising model for the attack, we sample an image $x_0$ from the training dataset, uniformly select a time step $t$ from 1 to $T$, sample a noise $\epsilon$ from $\mathcal{N}(0, I)$, and then derive the noisy image $x_t$ using $x_0$ and the sampled noise, while simultaneously querying the target model $M_1$ to obtain the intermediate feature map $y$:

$$x_t = \sqrt{\gamma_t}x_0 + \sqrt{1-\gamma_t}\epsilon, \tag{4}$$
$$y = M_1(x_0). \tag{5}$$

---

**Algorithm 1** Training

**while** *not converged* **do**
  $x_0 \sim q(x_0)$
  $t \sim \text{Uniform}(\{1, ..., T\})$
  $\epsilon \sim N(0, I)$
  $y = M_1(x_0)$
  Optimize a step on
  $\nabla_\theta \| f_{\theta_2}(\sqrt{\gamma_t}x_0 + \sqrt{1-\gamma_t}\epsilon, f_{\theta_1}(y), t) - \epsilon \|^2$
**end while**

---

Figure 2 provides a visualization of the proposed conditional denoising model, comprising an encoder $f_{\theta_1}$ and a denoising model $f_{\theta_2}$. The $f_{\theta_2}$ component is constructed based on a UNet architecture and a companion network. The training process is to train the complete conditional denoising model $f_\theta$ to predict the introduced noise. This training aims to minimize the objective function:

$$\| f_{\theta_2}(x_t, f_{\theta_1}(y), t) - \epsilon \|^2, \tag{6}$$

where $\epsilon \sim \mathcal{N}(0, I)$ represents the added noise at time step $t$.

#### 3.3.2 INFERENCE AND ATTACK

The inference and attack process is shown in Algorithm 2, it aims to reconstruct the input $x_0$ based on the intermediate feature map $y$. This process involves a reverse Markovian process, initiated with Gaussian noise $x_T \sim \mathcal{N}(0, I)$:

$$p_\theta(x_{0:T}|y) = p(x_T)\Pi_{t=1}^{T} p_\theta(x_{t-1}|x_t, y), \tag{7}$$

$$p(x_T) = \mathcal{N}(x_T|0, I), \tag{8}$$

$$p_\theta(x_{t-1}|x_t, y) = \mathcal{N}(x_{t-1}|\mu_\theta(x_t, y, t), \sigma_t^2 I). \tag{9}$$

We define the inference process as a Gaussian conditional distribution shown in Equation 11. Given that the noise introduced at time step $t$ can be approximated by $f_\theta$, similar to (Ho et al., 2020), we can parameterize the mean of $p_\theta(x_{t-1}|x_t, y)$ and $x_{t-1}$ as follows:

---

**Algorithm 2** Inference and Attack

$x_T \sim N(0, I)$
**for** $t = T, ..., 1$ **do**
  $\epsilon \sim N(0, I)$ if $t > 1$, else $\epsilon = 0$
  $x_{t-1} = \frac{1}{\sqrt{1-\beta_t}}(x_t - \frac{\beta_t}{\sqrt{1-\gamma_t}}f_{\theta_2}(x_t, f_{\theta_1}(y), t)) + \sqrt{\beta_t}\epsilon$
**end for**
return $x_0$

---

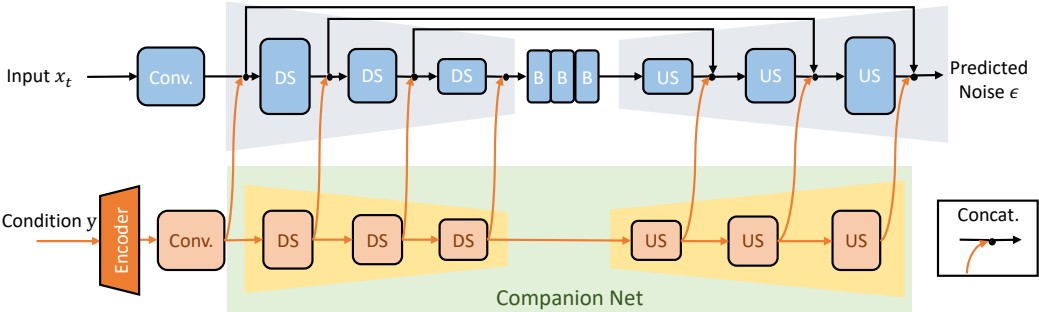

Figure 2: Architecture of the proposed diffusion-based inverse network (DS: downsampling layer; B: bottom convolutional layer; US: upsampling layer)

$$\mu_\theta(x_t, y, t) = \frac{1}{\sqrt{1-\beta_t}}\left(x_t - \frac{\beta_t}{\sqrt{1-\gamma_t}}f_{\theta_2}(x_t, f_{\theta_1}(y), t)\right) \tag{10}$$

$$x_{t-1} \leftarrow \mu_\theta(x_t, y, t) + \sqrt{\beta_t}\epsilon \tag{11}$$

where $\epsilon \sim \mathcal{N}(0, I)$. The attack procedure iteratively employs the above parameterization over $T$ steps to derive the reconstructed $\hat{x}_0$.

### 3.4 FEATURE MAP AWARENESS CONDITIONING MECHANISM

To provide conditional guidance, the intermediate feature map $y$ is propagated through a lightweight encoder network $f_{\theta_1}$. This encoder network performs parameterized reshaping of $y$ to match the dimensions of $x_t$, thereby simplifying the incorporation of conditioning information.

A straightforward method of incorporating conditional guidance involves concatenating the encoder output $\hat{y}$ with $x_t$ along the channel dimension. However, using this method often results in evident artifacts that significantly impact the quality of the reconstructed images. We hypothesize this drawback arises from the fact that the conditional information primarily guides the process at the outset and diminishes in significance as data traverses the UNet, particularly during the upsampling phase. To address this concern, we introduce a novel feature map awareness conditioning mechanism that integrates a companion network into $f_{\theta_2}$. As shown in Figure 2, the bottom branch corresponds to the companion network, which only accepts and processes input from the condition which is the intermediate feature map. Each downsampling or upsampling layer in the UNet has a corresponding companion layer in the companion network, and the output of each companion layer is concatenated with the output of the corresponding layer in the UNet along the channel dimension. This companion network provides consistent conditional guidance throughout the propagation process, especially during the upsampling phases. This ensures that the reconstructed image adheres to the condition of the intermediate feature map. Empirical results presented in Section 4.5 demonstrate the effectiveness of the feature map awareness conditioning mechanism compared to the alternative conditioning method.

## 4 EXPERIMENTS

### 4.1 MODEL ARCHITECTURE

To implement the attack model, we utilize an encoder $f_{\theta_1}$ with several transpose convolution layers to align with the dimensions of $x_0$ and one ResNet basic block (He et al., 2016). Each transpose layer doubles the size of the condition and the quantity of transpose layers varies based on the dimension of the intermediate feature map. For the UNet $f_{\theta_2}$, we employ three downsampling layers, three middle convolution layers, and three upsampling layers. After each downsampling or upsampling layer, we incorporate a self-attention layer. As for the companion network, we use three companion downsampling layers and three companion upsampling layers.

Table 1: Dataset and Model Information

| Attack Setting | Target Model Dataset | Attack Model Dataset | Resolution | Attack Model Training Data Size | Target Model | Spilit | Feature Dim. |
|---|---|---|---|---|---|---|---|
| Same Dataset | CIFAR10 | CIFAR10 | $32 \times 32$ | $40k$ | CNN | ReLU22 ReLU32 | (B, 128, 16, 16) (B, 128, 8, 8) |
| | | | | | ResNet | Layer1 | (B, 64, 32, 32) |
| | | | | | | Layer2 | (B, 128, 16, 16) |
| Diff. Dataset | CIFAR10 | Tiny-ImageNet | $32 \times 32$ | $40k$ | ViT[*] | Block3 Block6 Block9 | (B, 65, 192) (B, 65, 192) (B, 65, 192) |

[*] ViT outputs are shaped like (B, N, C), in which B is the batch size, N is the concatenation of patches and one class embedding, and C is the hidden dimension. The output will be mapped to (B, N-1, C) and then resized to (B, C, $\sqrt{N-1}$, $\sqrt{N-1}$.)

## 4.2 EXPERIMENTAL SETUP

In the experiments, the target model is trained using the CIFAR-10 (Krizhevsky et al., 2009) training dataset. The evaluation of the attack is performed on the CIFAR-10 test set, and the metrics are averaged over this set. Followed (He et al., 2019), to assess the effectiveness of the proposed attack in various scenarios, we train the attack models under two distinct settings:

- **Same Dataset:** For this setting, the attack model is trained on a subset of the CIFAR-10 training dataset with 40k images.

- **Different Dataset:** In this setting, the attack model's training data is distinct from the target model's training set. Specifically, we employ the Tiny-ImageNet dataset (Le & Yang, 2015) as the training set for the attack model and CIFAR-10 for training the target model. To circumvent overlap and potential bias, we excluded categories (35 in total) that might correlate with CIFAR-10. From the remaining 165 categories, 40k images are randomly selected for training the attack models.

To evaluate the proposed attack, we selected three distinct models as targets for our attacks:

- **CNN:** This model is a Convolutional Neural Network comprising six convolutional layers and two fully connected layers. The CNN model is partitioned at the end of the fourth (ReLU22) and sixth (ReLU32) convolutional layers.

- **ResNet:** We utilize the ResNet-18 (He et al., 2016) architecture from the torchvision library. To accommodate the dataset, we modified the initial convolutional layer and the fully connected layer. Divisions are made at the end of the first, second, and third basic blocks.

- **ViT:** We employ the Data-efficient Image Transformer (DeiT) tiny model configured with three heads and 12 attention blocks (Touvron et al., 2021). The patch size is adapted to be compatible with the dimensions of the input images. The ViT model is segmented at the end of the third, sixth, and ninth attention blocks.

The above dataset and model information are also presented in Table 1. We train the attack model for 500 epochs and use a learning rate of 3e-6 and batch size of 32. For the hyperparameter of the diffusion process, we use $T = 1000$ diffusion steps and the noise variance $\beta_t$ is uniformly sampled between 0.0001 and 0.02. All attacks take less than 1 minute. Experiments were conducted utilizing NVIDIA Tesla A40 (48GB) and RTX A6000 GPUs.

## 4.3 COMPARISON WITH PRIOR-ART ON CNN AND RESNET

In this section and section 4.4, we compare the proposed method with previous approaches, namely, the black-box attack (BBA) in (He et al., 2019) and enhanced inverse-network attack (EINA) (Li et al., 2022).

In Figure 1, we present visualizations comparing reconstructed images generated by previous approaches and our proposed method. Ground truth images are included for reference. These reconstructed inputs are based on randomly selected images. It can be observed that the prior approaches

Table 2: Performance comparison with prior attacks at two split points for CNN

| Metric | Setting | BBA He et al. (2019) | | EINA Li et al. (2022) | | DIA (ours) | |
|---|---|---|---|---|---|---|---|
| | | ReLU22 | ReLU32 | ReLU22 | ReLU32 | ReLU22 | ReLU32 |
| SSIM ↑ | | 0.6851 | 0.4349 | 0.7335 | 0.4353 | **0.9392** | **0.5041** |
| PSNR ↑ | Same Set | 23.38 | 19.50 | 24.31 | 19.63 | **31.77** | **19.84** |
| MSE ↓ | | 0.1047 | 0.2559 | 0.0846 | 0.2482 | **0.0153** | **0.2367** |
| SSIM ↑ | | 0.6358 | 0.3562 | 0.7306 | 0.4176 | **0.9336** | **0.4762** |
| PSNR ↑ | Diff Set | 22.17 | 18.37 | 24.30 | 19.37 | **31.42** | **19.50** |
| MSE ↓ | | 0.1383 | 0.3315 | 0.0847 | 0.2630 | **0.0166** | **0.2559** |

Table 3: Performance comparison with prior attacks at two split points for ResNet

| Metric | Setting | BBA He et al. (2019) | | EINA Li et al. (2022) | | DIA (ours) | |
|---|---|---|---|---|---|---|---|
| | | Layer1 | Layer2 | Layer1 | Layer2 | Layer1 | Layer2 |
| SSIM ↑ | | 0.7613 | 0.6377 | 0.7641 | 0.7373 | **0.9041** | **0.8411** |
| PSNR ↑ | Same Set | 24.67 | 22.53 | 24.94 | 24.50 | **29.07** | **28.30** |
| MSE ↓ | | 0.0777 | 0.1271 | 0.0731 | 0.0808 | **0.0158** | **0.0341** |
| SSIM ↑ | | 0.7622 | 0.5812 | 0.7699 | 0.7378 | **0.8711** | **0.7810** |
| PSNR ↑ | Diff Set | 24.66 | 21.72 | 25.02 | 24.53 | **28.79** | **25.91** |
| MSE ↓ | | 0.0778 | 0.1531 | 0.0718 | 0.0804 | **0.0306** | **0.0586** |

demonstrate similar reconstructed results, while our method aligns significantly closer to the reference images, presenting reduced blur.

In Tables 2, 3, we present the results on two target models, CNN and ResNet. We employ SSIM and PSNR metrics to assess the quality of the reconstructed images, complemented by MSE to measure the error. The results indicate that the proposed attack performs well at shallow split points of CNN and ResNet. As the split point deepens, the reconstruction quality of all attacks on CNN and ResNet decreases.

While training the attack models on a different dataset exhibits marginal variance in results at shallow split points, this variance becomes more pronounced with a deeper split point. Nonetheless, in comparison to previous methods, our attack consistently outperforms in all tested scenarios across all three metrics. More precisely, the proposed attack achieves an average improvement (across prior attacks and split points) of $29\%$, $20\%$, $30\%$ in terms of SSIM, PSNR, and MSE when applied to convolutional neural networks (CNN) and $18\%$, $17\%$, $61\%$ to ResNet models.

## 4.4 COMPARISON WITH PRIOR-ART ON VIT AND VULNERABILITY OF VIT

In Table 4, we present the results of the proposed method and compare it with previous approaches on ViT. It can be observed that our attack outperforms prior methods on ViT in all tested scenarios across all three metrics, demonstrating an average improvement (across prior attacks and split points) of $55\%$, $54\%$, $84\%$.

Notably, when compared to CNN and ResNet, the reconstructions on ViT models maintain consistently higher quality across all split points and the advantage of our approach over earlier attacks is more significant. We analyze and attribute this result to two potential reasons. While the behavior of downsampling is common in CNNs, the ViT maintains a similar tensor dimension and lacks this process. The absence of downsampling could lead to intermediate features retaining a higher fidelity to the original input image details Yin et al. (2023), rendering the ViT more susceptible to attacks. Moreover, in ViT, non-linear layers are not as frequent as in CNNs and ResNet. Several studies (He et al., 2019; Zhang et al., 2023; Yin et al., 2023) have shown that the presence of non-linear layers

Table 4: Performance comparison with prior attacks at three split points for ViT

| Metric | Setting | BBA He et al. (2019) | | | EINA Li et al. (2022) | | | DIA (ours) | | |
|---|---|---|---|---|---|---|---|---|---|---|
| | | Block3 | Block6 | Block9 | Block3 | Block6 | Block9 | Block3 | Block6 | Block9 |
| SSIM ↑ | Same Set | 0.5480 | 0.5433 | 0.5314 | 0.7651 | 0.7620 | 0.7652 | **0.9845** | **0.9657** | **0.9552** |
| PSNR ↑ | | 21.43 | 21.36 | 21.26 | 24.95 | 24.80 | 24.96 | **40.64** | **27.78** | **27.93** |
| MSE ↓ | | 0.1645 | 0.1668 | 0.1709 | 0.0730 | 0.0756 | 0.0728 | **0.0020** | **0.0414** | **0.0448** |
| SSIM ↑ | Diff Set | 0.5276 | 0.5321 | 0.5087 | 0.7648 | 0.7645 | 0.7650 | **0.9805** | **0.9742** | **0.9617** |
| PSNR ↑ | | 21.22 | 21.37 | 21.03 | 24.95 | 24.93 | 24.95 | **39.69** | **38.58** | **37.41** |
| MSE ↓ | | 0.1722 | 0.1665 | 0.1806 | 0.0730 | 0.0732 | 0.0729 | **0.0024** | **0.0032** | **0.0042** |

not only enhances model generalization but also obfuscates some model details, possibly increasing the model's resilience against inverse network attacks.

## 4.5 IMPORTANCE OF THE FEATURE MAP AWARENESS CONDITIONING MECHANISM

To evaluate the significance of the proposed feature map awareness conditioning mechanism utilizing the companion network in the attack, we replace it with a prevalent condition mechanism (Rombach et al., 2022) that employs cross-attention layers to incorporate condition information into the UNet. The attack results are presented in Tables 5, 6, 7. The results indicate that the feature map awareness conditioning mechanism is significantly more effective than the cross-attention mechanism and more suitable for inverse network attacks.

Table 5: Performance comparison of conditioning mechanism on CNN

| Metric | Setting | Cross Attention | | DIA (ours) | |
|---|---|---|---|---|---|
| | | ReLU22 | ReLU32 | ReLU22 | ReLU32 |
| SSIM ↑ | Same Set | 0.5132 | 0.3524 | **0.9392** | **0.5041** |
| PSNR ↑ | | 20.34 | 17.58 | **31.77** | **19.84** |
| MSE ↓ | | 0.2110 | 0.3983 | **0.0153** | **0.2367** |
| SSIM ↑ | Diff Set | 0.2098 | 0.3012 | **0.9336** | **0.4762** |
| PSNR ↑ | | 16.48 | 16.92 | **31.42** | **19.50** |
| MSE ↓ | | 0.5143 | 0.4641 | **0.0166** | **0.2559** |

Table 6: Performance comparison of conditioning mechanism on ResNet

| Metric | Setting | Cross Attention | | DIA (ours) | |
|---|---|---|---|---|---|
| | | Layer1 | Layer2 | Layer1 | Layer2 |
| SSIM ↑ | Same Set | 0.1504 | 0.1669 | **0.9041** | **0.8411** |
| PSNR ↑ | | 15.67 | 15.50 | **29.07** | **28.30** |
| MSE ↓ | | 0.6140 | 0.6424 | **0.0158** | **0.0341** |
| SSIM ↑ | Diff Set | 0.1343 | 0.1365 | **0.8711** | **0.7810** |
| PSNR ↑ | | 15.67 | 15.36 | **28.79** | **25.91** |
| MSE ↓ | | 0.6207 | 0.6646 | **0.0306** | **0.0586** |

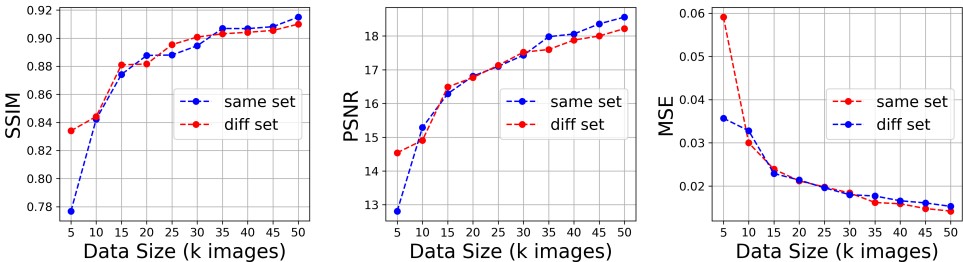

Figure 3: Impact of training data size on the SSIM, PSNR, and MSE for the proposed attack on ReLU22 of CNN

Table 7: Performance comparison of conditioning mechanism on ViT

| Metric | Setting | Cross Attention | | | DIA (ours) | | |
|---|---|---|---|---|---|---|---|
| | | Block3 | Block6 | Block9 | Block3 | Block6 | Block9 |
| SSIM ↑ | | 0.1881 | 0.1856 | 0.1888 | **0.9845** | **0.9657** | **0.9552** |
| PSNR ↑ | Same Set | 15.74 | 15.81 | 15.89 | **40.64** | **27.78** | **27.93** |
| MSE ↓ | | 0.6107 | 0.6019 | 0.5892 | **0.0020** | **0.0414** | **0.0448** |
| SSIM ↑ | | 0.1851 | 0.1784 | 0.1679 | **0.9805** | **0.9742** | **0.9617** |
| PSNR ↑ | Diff Set | 15.92 | 15.76 | 15.44 | **39.69** | **38.58** | **37.41** |
| MSE ↓ | | 0.5851 | 0.6072 | 0.6532 | **0.0024** | **0.0032** | **0.0042** |

## 4.6 IMPACT OF DATA SIZE ON THE PROPOSED ATTACK

In this section, we analyze the impact of training data size on the proposed attack. We select the CNN as our target model, partitioning it at the fourth layer ReLU22, and evaluate varying training data sizes under both the same set and different set configurations. As shown in Figure 3, the results demonstrate a positive correlation between training data size and the effectiveness of the attack. As the dataset size increases, SSIM values ascend and plateau at peak levels. Simultaneously, PSNR exhibits a steady rise, while MSE values steadily decline.

## 5 CONCLUSIONS AND FUTURE WORK

In this paper, we present a diffusion-based inverse network attack on collaborative inference systems. Our attack leverages a novel feature map awareness conditioning mechanism that utilizes a companion network tailored for inverse network attacks. Our extensive experiments demonstrate the effectiveness of the proposed attack, surpassing prior approaches across three target models. To the best of our knowledge, the proposed attack sets a new state of the art in attacking collaborative inference systems.

Moreover, our results reveal a significant vulnerability of the collaborative inference of ViT models. In the future, as computational resources allow, we will explore whether there is a similar vulnerability for larger transformer-based models, such as large language models including GPT (OpenAI, 2023) and Llama 2 (Touvron et al., 2023). Given the broader applications of transformer-based models, we will explore defensive measures to enhance their resilience.

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
