# DIA: Diffusion based Inverse Network Attack on Collaborative Inference

## A  Appendix

### A.1  More Random examples of Reconstructed Inputs from DIA

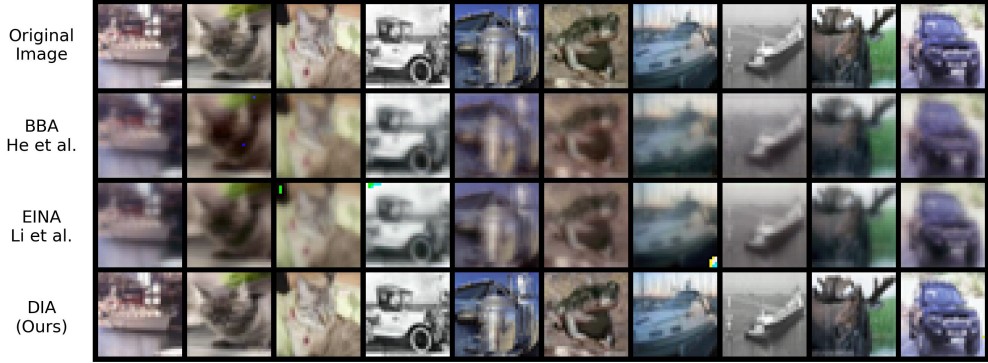

Figure 1: Reconstructed inputs comparison of prior approaches and our method on Layer1 of ResNet

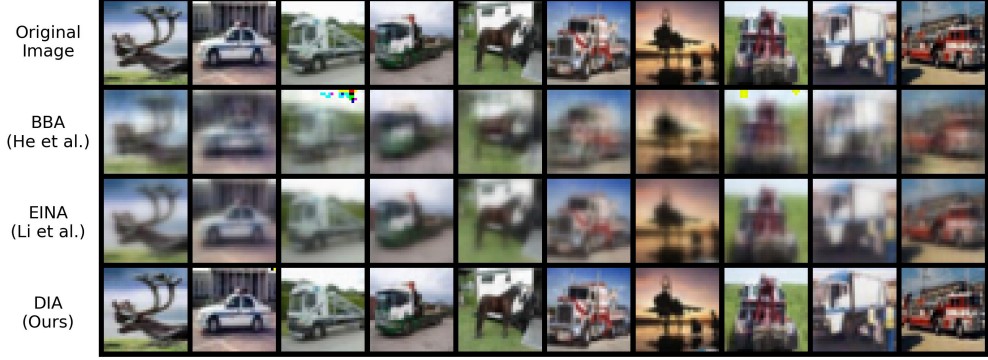

Figure 2: Reconstructed inputs comparison of prior approaches and our method on Block3 of ViT