# OpenReview forum: "DIA: Diffusion based Inverse Network Attack on Collaborative Inference"
_ICLR.cc/2024/Conference — ICLR 2024 Conference Withdrawn Submission_

### Official Review · Reviewer_wMLT · 2023-10-24

**Soundness:** 3 good
**Presentation:** 2 fair
**Contribution:** 2 fair
**Rating:** 3
**Confidence:** 4

**Summary:**

The paper introduces a diffusion-based inverse network attack called DIA for collaborative inference systems. DIA employs a feature map awareness conditioning mechanism and outperforms previous attacks in terms of SSIM, PSNR, and MSE, particularly on CNN, ResNet, and ViTs. The study identifies the vulnerability of ViTs in collaborative inference, raising concerns about deploying transformer-based models in such systems. The paper presents improved reconstruction results and insights into the vulnerability of ViT models.

**Strengths:**

- The proposed feature map awareness conditioning mechanism appears to be novel to best of my knowledge
- The authors present superior performance to the compared methods
- The paper is easy to follow.

**Weaknesses:**

- The results in Figure 1 for BBA and EINA look relatively weak. The literature [1] indicates that it shouldn't be an issue to retrieve the original image from intermediate layers. Especially for relatively early splits.
- The authors did not compare previous methods for Table 5,6,7. Why not?
- The distribution of the "different" datasets seems to remain close (CIFAR 10, Tiny ImageNet). Could the authors choose more disjoint datasets, such as Tiny ImageNet and FairFace (or similar)?
- Could the authors also compare the retrieval performance of a simple autoencoder network?
- The paper misses ablation studies to justify the design choices behind the awareness conditioning mechanism.

[1] Noisy adversarial representation learning for effective and efficient image obfuscation; UAI 2023

**Questions:**

Please address the points mentioned in the weakness section.

---

### Official Review · Reviewer_9V3L · 2023-10-28

**Soundness:** 3 good
**Presentation:** 2 fair
**Contribution:** 2 fair
**Rating:** 5
**Confidence:** 4

**Summary:**

This paper introduces a novel diffusion-based inverse network attack (DIA) designed for collaborative inference systems, emphasizing the increasing importance of evaluating privacy and security in resource-constrained computing environments where large neural networks are deployed. The study provides extensive empirical results, demonstrating the effectiveness of DIA in significantly improving image quality metrics when applied to various neural network architectures, particularly Vision transformers (ViTs), highlighting their substantial vulnerability. As a result, the paper raises a red flag regarding the deployment of transformer-based models in collaborative inference systems, underscoring the necessity for heightened security considerations in such settings.

**Strengths:**

1.	The paper highlights the critical issue of evaluating the privacy and security aspects of collaborative inference systems, suggesting that as these systems become more widespread, understanding their potential vulnerabilities is essential.
2.	The paper introduces a novel attack method called "DIA" (Diffusion-based Inverse Network Attack) designed specifically for collaborative inference systems. DIA leverages a unique feature map awareness conditioning mechanism to guide the diffusion model.

**Weaknesses:**

The paper provides an overview of the DIA attack but lacks in-depth methodological details. Readers may find it challenging to understand the attack's intricacies, as the paper does not delve deeply into the specifics of how the attack works. A more detailed explanation of the attack mechanism, algorithms, and technical intricacies would enhance the paper's comprehensibility and its potential for replication by other researchers.

**Questions:**

1.	In section 3.2, do the authors contemplate a situation where the server needs to make multiple requests to the client for acquiring the intermediary feature output to train the inversion model? Is this assumption feasible?
2.	The process of training high-performance diffusion models can be costly. What is the additional training burden associated with the Companion Net?
3.	The Companion Net directs the diffusion model using channel concatenation, resembling established image guidance techniques (where the lightweight network functions akin to the CLIP image encoder). Could you provide a more detailed explanation of the significance of this method?

---

### Official Review · Reviewer_ZM8C · 2023-11-01

**Soundness:** 2 fair
**Presentation:** 2 fair
**Contribution:** 2 fair
**Rating:** 3
**Confidence:** 3

**Summary:**

The paper delves into Inverse Network Attacks in Deep Learning, mainly focusing on collaborative inference systems within resource-constrained computing environments. A significant highlight of the study is the introduction of a "diffusion-based inverse network attack," abbreviated as DIA, designed specifically for the image task in collaborative inference systems. This technique innovates with a feature map awareness conditioning mechanism that steers the diffusion model. The paper sets itself apart by presenting empirical results that showcase the proposed attack's efficacy, indicating substantial improvement percentages with different deep learning networks compared to previous methodologies. The paper concludes experimentally that the ViT model is more vulnerable.

**Strengths:**

1.	Scenario: The paper proposes a critical scenario, the collaborative inference system within a resource-constrained computing environment. As deep learning models get larger (e.g. LLMs), collaborative inference would be common. Therefore, exploring attacks in this scenario is necessary.
2.	System: The paper provides a complete system, including the training and inference details.
3.	Novelty: The paper has a Feature Map Awareness Conditioning Mechanism, which is an extra companion net added on a Unet. By concatenating the intermediate outputs of Unet with those of the Companion Net, the model can predict the inputs of the target model more accurately.

**Weaknesses:**

1.	Dataset: The paper provides two different scenarios for the experiments: the same dataset and a different dataset. However, both scenarios use the same image size. It is too ideal to assume the target model and attack model have the same input image size. Moreover, the resolution of an image is 32 * 32, which is tiny and limited performance for a realistic environment.
2.	Target Model: The paper provides three target models: CNN, ResNet, and ViT. The model size of the target models is tiny, especially the CNN. It only has six convolutional layers and two fully connected layers. Commonly, it does not need a collaborative inference system to handle. The author mentions the LLMs and the resource-constrained computing environments in the introduction, which is inconsistent with the target model size. Using this attack on such a small model is not convincing.
3.	Vulnerability of ViT: The paper uses an empirical method to evaluate the vulnerability of ViT. However, the paper uses a tiny ViT, which only has three heads and 12 attention blocks. It cannot represent all ViT-based models.

**Questions:**

please respond to the weaknesses